# Shp2 in uterine stromal cells critically regulates on time embryo implantation and stromal decidualization by multiple pathways during early pregnancy

Jianghong Cheng[1], Jia Liang[1], Yingzhe Li[1], Xia Gao[1], Mengjun Ji[1], Mengying Liu[1], Yingpu Tian[1], Gensheng Feng[2], Wenbo Deng[3,4], Haibin Wang[3,4]*, Shuangbo Kong[3,4]*, Zhongxian Lu[1,4]*

1 School of Pharmaceutical Sciences, State Key Laboratory of Cellular Stress Biology, Xiamen University, Xiamen, Fujian, China, 2 Department of Pathology, Division of Biological Sciences, University of California San Diego, La Jolla, California, United States of America, 3 Reproductive Medical Centre, The First Affiliated Hospital of Xiamen University, Xiamen, Fujian, China, 4 Fujian Provincial Key Laboratory of Reproductive Health Research, Medical College of Xiamen University, Xiamen, Fujian, China

☯ These authors contributed equally to this work.
* haibin.wang@vip.163.com (HW); kongshb1313@gmail.com (SK); zhongxian@xmu.edu.cn (ZL)

**Data Availability Statement:** All relevant data are within the manuscript and its Supporting Information files.

## Abstract

Approximately 75% of failed pregnancies are considered to be due to embryo implantation failure or defects. Nevertheless, the explicit signaling mechanisms governing this process have not yet been elucidated. Here, we found that conditional deletion of the Shp2 gene in mouse uterine stromal cells deferred embryo implantation and inhibited the decidualization of stromal cells, which led to embryonic developmental delay and to the death of numerous embryos mid-gestation, ultimately reducing female fertility. The absence of Shp2 in stromal cells increased the proliferation of endometrial epithelial cells, thereby disturbing endometrial epithelial remodeling. However, Shp2 deletion impaired the proliferation and polyploidization of stromal cells, which are distinct characteristics of decidualization. In human endometrial stromal cells (hESCs), Shp2 expression gradually increased during the decidualization process. Knockout of Shp2 blocked the decidual differentiation of hESCs, while Shp2 overexpression had the opposite effect. Shp2 knockout inhibited the proliferation of hESCs during decidualization. Whole gene expression profiling analysis of hESCs during the decidualization process showed that Shp2 deficiency disrupted many signaling transduction pathways and gene expression. Analyses of hESCs and mouse uterine tissues confirmed that the signaling pathways extracellular regulated protein kinases (ERK), protein kinase B (AKT), signal transducer and activator of transcription 3 (STAT3) and their downstream transcription factors CCAAT/enhancer binding protein β (C/EBPβ) and Forkhead box transcription factor O1 (FOXO-1) were involved in the Shp2 regulation of decidualization. In summary, these results demonstrate that Shp2 plays a crucial role in stromal decidualization by mediating and coordinating multiple signaling pathways in uterine stromal cells. Our discovery possibly provides a novel key regulator of embryo implantation and novel therapeutic target for pregnancy failure.

**Funding:** This work was supported by the National Key R&D program of China (2017YFC1001402 to Z.L. and H.W., 2018YFC1003701 to Z.L., 2018YFC1004404 to S.K.) and the National Natural Science Foundation of China (31671564 to Z.L., 31970797 to Z.L). The funders had no role in the study design, data collection and analysis, decision to publish, or preparation of the manuscript.

**Competing interests:** The authors have declared that no competing interests exist.

## Author summary

Embryo implantation includes the establishment of uterine receptivity, blastocyst attachment, and endometrial decidualization. Disorders of this process usually induce pregnancy failure, resulting in women infertility. But the signaling mechanisms governing this process remain unclear. Here, using gene knockout mouse model and human endometrial stromal cells (hESCs), we identified a novel key regulator of embryo implantation, Shp2, which plays a crucial role in stromal decidualization by mediating and coordinating multiple signaling pathways in uterine stromal cells. Shp2 deficiency in mouse uterine stromal cells inhibited the uterine stromal decidualization, disturbing embryo implantation and embryonic development, ultimately reducing female fertility. The absence of Shp2 in hESCs also blocked the decidual differentiation. Our findings not only promote the understanding of peri-implantation biology, but may also provide a critical target for more effectively diagnose and/or treat women with recurrent implantation failure or early pregnancy loss.

## Introduction

In recent years, infertility has gradually increased, causing serious health and social problems, and disorders of early pregnancy are one of the main reasons [1]. Early pregnancy includes the establishment of uterine receptivity, blastocyst attachment, and endometrial decidualization [2–4]. Defects in these processes can produce adverse ripple effects throughout pregnancy, leading to compromised pregnancy outcomes, including infertility, miscarriages, preterm birth and preeclampsia [2,5,6]. Embryo implantation is comprehensively regulated by a variety of regulatory factors led by estrogen and progesterone, but the complex dynamic signaling network is still poorly understood [3,5].

Embryo implantation involves drastic morphological and functional changes in endometrial epithelial cells and stromal cells [2,3,7,8]. In mice, uterine epithelial cells proliferate in response to the first wave of estrogen on the eve of ovulation, but their proliferation is stopped by progesterone prior to embryo attachment [8,9]. At the same time, progesterone induces the differentiation and remodeling of epithelial cells, stimulates the proliferation of stromal cells, and subsequently induces the decidual differentiation of stromal cells [3,7,8]. The biological effects of estrogen and progesterone are mediated by their nuclear receptors and by a variety of paracrine factors, including prostaglandins, Indian hedgehog (IHH), leukemia inhibitory factor (LIF), bone morphogenetic protein-2 (BMP2), wingless type MMTV integration site family member 4 (Wnt4), insulin-like growth factor 1 (IGF), and epidermal growth factor (EGF) [8,10–16]. Environmental factors, including inflammation, stress, and metabolic disorders, can affect embryo implantation by interfering with these regulatory signals [1,5,7]. These signals activate a variety of intracellular signaling pathways, such as adenosine-3',5'-cyclic adenosine phosphate (cAMP)-protein kinase A (PKA), mitogen-activated kinase (MAPK)/ERK, phosphatidylinositol 3-kinase (PI3K)-AKT, and JAK-Stat3, which are coordinated and integrated with the estrogen receptor (ER) and progesterone receptor (PR) signaling pathways to exert biological effects on uterine cell proliferation and differentiation [17–22]. However, the basic mechanisms of integration and coordination among signaling pathways remain unclear, and the key regulatory proteins in this signaling network are not well known [1,2,5].

Encoded by the *PTPN11* gene, Shp2 is a ubiquitously expressed protein tyrosine phosphatase (PTP) [23] that transduces and integrates signaling pathways (such as MAPK/ERK,

PI3K-AKT, and JAK-Stat3) triggered by growth factors, cytokines, hormones, and antigens during cell growth, differentiation, migration, and death [24–26] and plays major roles in multiple physiological and pathological processes [27]. Dysregulation of Shp2 signaling (abnormal protein expression or genetic mutation) is involved in Noonan syndrome, Leopard syndrome, metabolic disorders, and several types of cancer, including breast cancer, liver cancer, and leukemia [28–32]. Shp2 is thought to be a potential therapeutic target [27,33]. Recently, the role of Shp2 in reproduction has begun to be uncovered. We and other groups have established that Shp2 plays an indispensable role in spermatogenesis by mediating follicle stimulating hormone (FSH) and testosterone signals [34–37]. Regarding the role of Shp2 in female reproduction, we recently established that the ablation of Shp2 in the uterus using PR-Cre results in complete infertility owing to the lack of embryo attachment to the uterine epithelium, implying that Shp2 serves as an indispensable signaling component in uterine biology [38]. Since embryo attachment is the trigger for stromal cell decidualization in mice, the PR-Cre/Shp2 mouse model cannot be used to explore the role of Shp2 in stromal cell differentiation.

Herein, we crossed Shp2$^{f/f}$ mice with Amhr2-Cre mice, which are widely used to delete target genes in the stroma and myometrium but not in uterine epithelial cells, to further identify the cell type-specific function of Shp2 in the uterus. The deletion of Shp2 in mesenchymal cells resulted in female subfertility owing to deferred embryo implantation and compromised decidualization. Using immortalized human endometrial stromal cells (hESCs), we confirmed the crucial role of Shp2 in decidualization and performed whole gene expression profiling analysis during decidualization. Multiple signals, including ERK, AKT, and STAT3, were found to be involved in the Shp2 regulation of decidualization. Our investigations further our understanding of the regulatory effects of Shp2 on female reproduction and provide clues for the treatment of common clinical diseases, including infertility and miscarriage.

## Results

### Stromal ablation of Shp2 impairs female fertility in mice

To investigate the physiological relevance of Shp2 in early pregnancy events, we first analyzed the uterine expression patterns of Shp2 in mice during early pregnancy. During implantation in mice, blastocyst trophectoderm cells first attach to the surface of the uterine luminal epithelium (LE) at approximately midnight (22:00–24:00) on the fourth day of pregnancy (D4) (D1 = see the vaginal plug) and then rapidly invade the underlying stromal compartment [4]. The Shp2 mRNA and protein expression levels were moderate in the endometrium on D4 and D5 but dramatically upregulated upon the initiation and progression of decidualization on D6 and beyond (**S1A and S1B Fig**). Immunohistochemistry staining revealed that Shp2 was mainly expressed in the LE and subluminal stromal cells on D4 and D5 (**S1C Fig,** S). Accompanied by the progression of decidualization, Shp2 was intensely localized in decidualizing cells (**S1C Fig,** DZ). This observation suggested that Shp2 is expressed in the peri-implantation uterus in a spatiotemporal manner, implying its potential involvement in decidual development postimplantation.

To explore the function of Shp2 in the stromal compartment, we used a Cre-LoxP approach to generate a conditional-knockout mouse model in which the gene expression of Shp2 was ablated specifically in Amhr2-expressing cells [39]. Mouse model harboring a specific deletion of Shp2 (Shp2$^{d/d}$) was generated by crossing Shp2-loxp mice (Shp2$^{f/f}$) with Amhr2-Cre mice (Amhr2$^{Cre/+}$). The uteri of Shp2$^{f/f}$ mice expressed the Shp2 protein in the LE and in subluminal stromal cells on D4 and in decidual cells on D6, whereas stromal cell staining was mostly absent in the uteri of Shp2$^{d/d}$ mice (**Fig 1A,** brown color). Consistently, the quantification of uterine Shp2 mRNA and protein expression levels also revealed significant reductions in

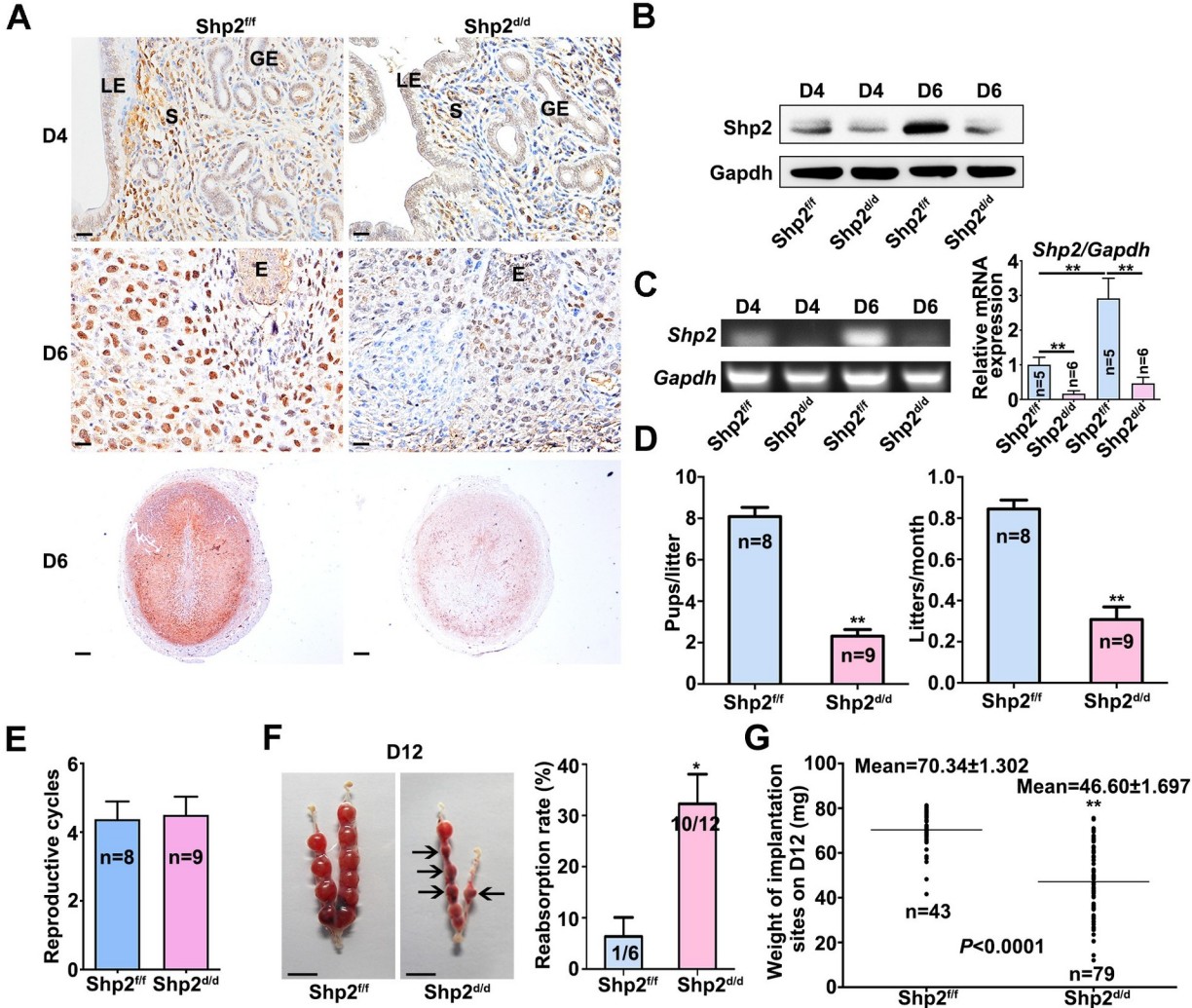

**Fig 1. Deletion of Shp2 from the uterine mesenchyme results in female subfertility. (A-B)** Shp2 protein levels were detected by immunohistochemistry (A) and western blot (B) in Shp2$^{f/f}$ (n = 6) and Shp2$^{d/d}$ (n = 7) D4 and D6 uteri. Gapdh served as the loading control. Brown color indicates positive staining. Tissues were counterstained with hematoxylin. Scale bars for the top 4 images in panel A, 20 μm; Scale bars for the bottom two images in panel A, 200 μm. LE, luminal epithelium. GE, glandular epithelium. S, stroma. DZ, decidual zone. E, embryo. **(C)** Detection of Shp2 mRNA in Shp2$^{f/f}$ (n = 6) and Shp2$^{d/d}$ (n = 7) D4 and D6 uteri by reverse transcription PCR. Quantitative analysis of Shp2 mRNA expression as showed in the right. **(D)** The fecundity of Shp2$^{f/f}$ (n = 8) and Shp2-deficient female mice (n = 9) was evaluated by analyzing their litter sizes and numbers of litters per month. The number within the bar indicates the number of mice examined. **(E)** Reproductive cycles between Shp2$^{f/f}$ (n = 8) and Shp2$^{d/d}$ (n = 9) female mice during 6-month breeding experiment. **(F)** Representative photographs of uteri and the reabsorption rates of embryos of Shp2$^{f/f}$ (n = 6) and Shp2$^{d/d}$ mice (n = 12) on D12. The arrows indicate degenerating and reabsorbing embryos. Scale bars, 1000 μm. The numbers within the bars indicate the number of mice with reabsorption sites per the total number of mice examined. **(G)** Implantation site weight of embryos of Shp2$^{f/f}$ (n = 6) and Shp2$^{d/d}$ mice (n = 12) on D12. The data are presented as mean ± SD from at least 6 mice in each group or three independent experiments. Statistical difference is indicated: $^*$P < 0.05; $^{**}$P < 0.01; $^{***}$P < 0.001.

Shp2$^{d/d}$ mice compared with Shp2$^{f/f}$ mice (**Fig 1B and 1C**). These results verified our successful deletion of Shp2 in uterine stromal cells.

Next, we evaluated the effect of Shp2 ablation on female fertility by conducting a 6-month breeding experiment. Shp2$^{f/f}$ and Shp2$^{d/d}$ females were placed in a cage at a 1:1 ratio with wild-type (WT) males of proven fertility and observed every 3 days for 6 months. The number of pups per litter and the total number of litters born were recorded. During the 6-month breeding period, the pups were separated immediately after weaning. Females of both genotypes

exhibited normal mating behavior, with the formation of vaginal plugs being observed after exposure to WT male mice. The Shp2f/f and Shp2d/d females had comparable reproductive cycles (approximately 4 cycles). However, the female Shp2d/d mice were subfertile, as evidenced by obviously reduced pup numbers per litter and litters per month (**Fig 1D**). Over a 6-month period, the Shp2$^{d/d}$ mice produced considerably fewer pups (average 8.44±2.32 pups/ mouse) than Shp2$^{f/f}$ mice (average 42.34±3.82 pups/mouse) (**Fig 1D**). No differences in reproductive cycles were found between Shp2$^{f/f}$ and Shp2$^{d/d}$ female mice (**Fig 1E**). Some embryos started to be reabsorbed, and their implantation site (IS) weights were strikingly decreased on day 12 of pregnancy compared with those of Shp2$^{f/f}$ mice (**Fig 1F and 1F,** arrow indicated).

Given that the Amhr2$^{Cre/+}$ mouse model can also drive conditional gene loss in ovarian surface epithelial and granulosa cells of developing follicles in adult ovaries [40], we next analyzed ovarian function in Shp2$^{d/d}$ mice. The ovaries of 8-week-old adult Shp2$^{d/d}$ mice appeared grossly and histologically normal (**S2A Fig**), and their capacities to respond to gonadotropins were similar in 3-week-old immature female mice, compared to the corresponding Shp2$^{f/f}$ mice (**S2B Fig**), which reflected by the similar oocytes production after the pharmacological induction of superovulation with pregnant mare serum gonadotropin and human chorionic gonadotropin in both Shp2$^{d/d}$ and Shp2$^{f/f}$ mice. We also analyzed ovarian steroid secretion levels and observed that the Shp2$^{d/d}$ mice exhibited no difference in circulating levels of progesterone and 17β-estradiol (E2) compared to Shp2$^{f/f}$ mice (**S2C and S2D Fig**) on D4. Our findings indicate that the subfertility phenotype of Shp2$^{d/d}$ female mice is not caused by ovarian function defects but rather likely due to uterine defects.

## Shp2 deficiency defers on-time implantation in mice

To ascertain the causes of subfertility, we first analyzed the implantation statuses of mice lacking uterine stromal Shp2 expression. All Shp2$^{f/f}$ mice (n = 6) exhibited distinct implantation sites at 2 o'clock (02:00) in the morning on day 5 of pregnancy, as examined by the intravenous injection of blue dye (**Fig 2A,** left image), whereas none of the Shp2$^{d/d}$ mice (n = 7) exhibited any signs of implantation (**Fig 2A,** right image). Morphologically normal blastocysts were recovered by flushing the Shp2$^{d/d}$ uteri (**Fig 2B**). However, all Shp2$^{d/d}$ mice exhibited blue bands when examined at 10:00 on day 5 of pregnancy (**Fig 2C**), although the blue bands in Shp2$^{d/d}$ uteri were not as obvious as those in the Shp2$^{f/f}$ group (**Fig 2D**). The results demonstrated that the loss of uterine stromal Shp2 defers on-time implantation.

In normal pregnant uteri, the receptive state is also marked by the cessation of epithelial cell proliferation before implantation [7]. As expected, Ki67, a cell proliferation marker, was undetectable in the uterine LEs of Shp2$^{f/f}$ mice on D4 of pregnancy (**Fig 2E**). In contrast, Shp2$^{d/d}$ uteri exhibited Ki67 expression in some LEs, which indicated sustained epithelial cell proliferation in the absence of Shp2 (**Fig 2E and 2F**). At the same time, the proliferation of endometrial stromal cells in Shp2$^{d/d}$ uteri notably reduced compared with that in the uteri of Shp2$^{f/f}$ mice (**Fig 2E and 2F**). Upon embryo attachment, luminal epithelial cells surrounding the blastocyst in the implantation chamber (crypt) disappear [41]. Inspection of the implantation sites on day 6 of pregnancy revealed that the luminal epithelial layer encompassing the implanting embryos was absent in Shp2$^{f/f}$ mice but still intact in Shp2$^{d/d}$ mice (**Fig 2G,** arrowhead indicated in the right image), as evidenced by the distinct immunostaining of cytokeratin (CK), a classical epithelial cell marker. In normal physiological conditions of mouse implantation, dynamic localization of Cox2 precisely correlates with different phases of the implantation process, and the localization of Cox2 shifts from the stroma surrounding the embryos in antimesometrial pole to the mesometrial pole between D5-D8 [42]. The abnormal localization of Cox2 in peri-implantation uterus may indicate the delayed embryo implantation [42]. In

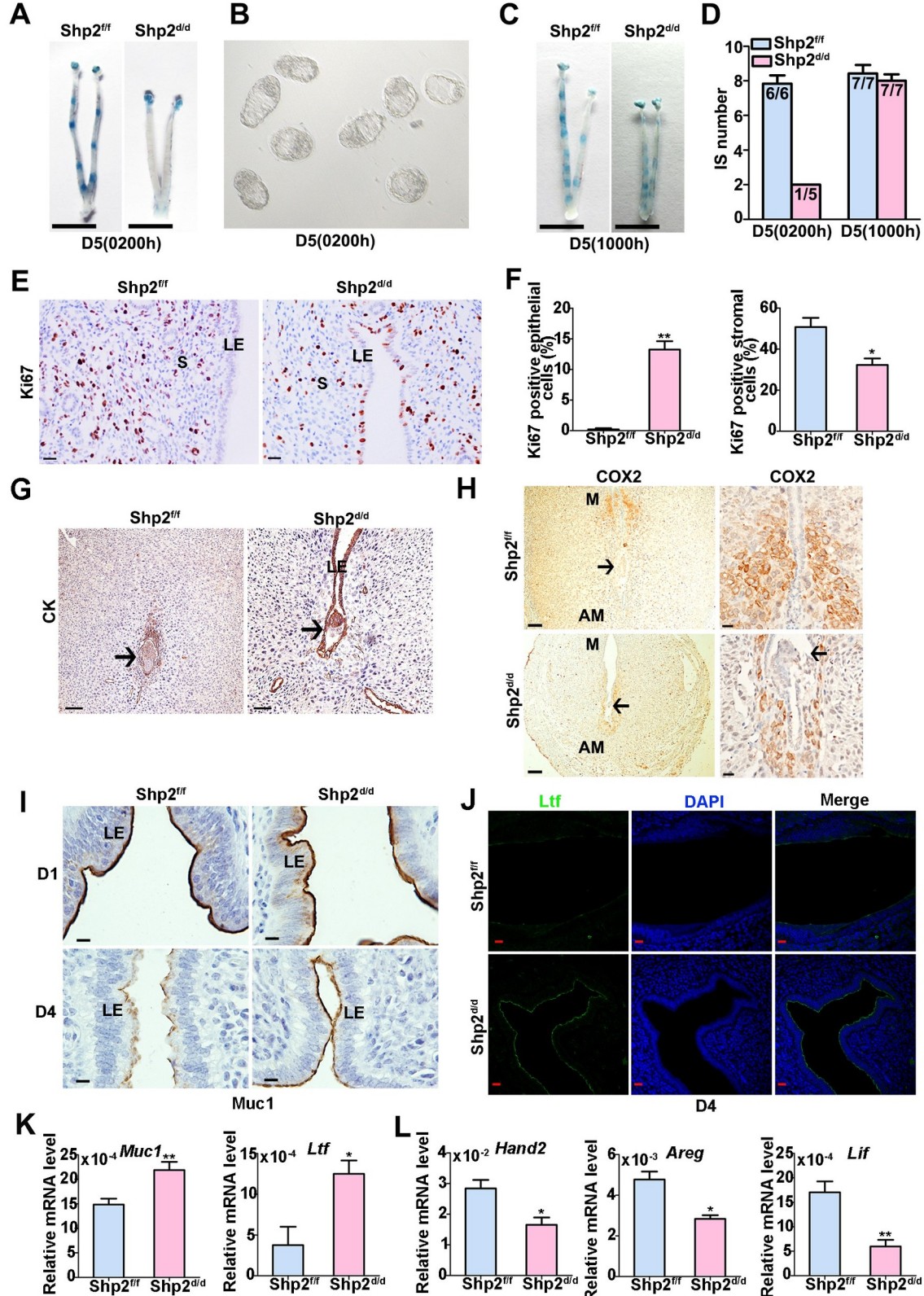

**Fig 2. Uterine mesenchymal deletion of Shp2 hampers on-time embryo implantation, disrupts uterine receptivity and defers blastocyst invasion in Shp2$^{d/d}$ mice. (A)** Representative photographs of Shp2$^{f/f}$ uteri (n = 6) with implantation sites (ISs) and Shp2$^{d/d}$

uteri (n = 7) without blue bands on day 5 (02:00). Scale bars, 1000 μm. **(B)** Representative photomicrographs of nonimplanted morphologically normal blastocysts recovered from Shp2$^{d/d}$ females (n = 7) without signs of a blue reaction at 02:00 on day 5. **(C)** Representative uteri of Shp2$^{f/f}$ (n = 7) and Shp2$^{d/d}$ mice (n = 7) at 10:00 on day 5. Mutant uteri exhibited weak blue bands. Scale bars, 1000 μm. **(D)** Number of ISs on day 5 (02:00 and 10:00). The numbers within the bars indicate the number of mice with an IS per the total number of mice examined. **(E)** Representative images of Ki67 expression in Shp2$^{f/f}$ (n = 6) and Shp2$^{d/d}$ (n = 6) uterine tissues on D4 as determined by immunohistochemistry staining. Scale bars, 20 μm. **(F)** Quantification of the Ki67-positive epithelial and stromal cells in (E). Six high power field per tissue were used to calculate the positive rate of Ki67. **(G)** Cytokeratin (epithelial marker) immunostaining of ISs collected from Shp2$^{f/f}$ (n = 6) and Shp2$^{d/d}$ (n = 6) mice on D6. Scale bar in the left panel, 100 μm. Scale bar in the right panel, 50 μm. **(H)** Cox2 immunostaining of ISs collected from Shp2$^{f/f}$ (n = 6) and Shp2$^{d/d}$ mice (n = 6) on D6. Scale bars in the left column, 100 μm; Scale bars in the right column, 20 μm. Brown color indicates positive staining. Tissues were counterstained with hematoxylin. M, mesometrial pole; AM, anti-mesometrial pole. **(I)** Representative image of Muc-1 expression in the uterine sections of Shp2$^{f/f}$ (n = 6) and Shp2$^{d/d}$ mice (n = 6) on D1 (upper panel) and D4 (lower panel) as determined by immunohistochemistry staining. Scale bars, 10 μm. **(J)** Lactoferrin (Ltf) expression in Shp2$^{f/f}$ (n = 6) and Shp2$^{d/d}$ (n = 6) D4 uteri as determined by immunofluorescence staining. The cell nuclei were stained with DAPI. Scale bars, 20 μm. **(K)** The mRNA levels of Muc-1 and Ltf in Shp2$^{f/f}$ (n = 6) and Shp2$^{d/d}$ (n = 6) D4 uteri as determined by quantitative PCR. **(L)** Relative expression levels of Hand2, Areg and Lif in Shp2$^{f/f}$ (n = 6) and Shp2$^{d/d}$ (n = 6) D4 uteri. LE, luminal epithelium. GE, glandular epithelium. S, stroma. DZ, decidual zone. E, embryo. The data are presented as the mean ± SD from at least 6 mice in each group or three independent experiments. Statistical differences are indicated as follows: $^*$P<0.05, $^{**}$P<0.01.

Shp2$^{f/f}$ mice, Cox2 correctly localized at the mesometrial pole on D6 (**Fig 2H,** arrowhead indicated in the top two images), in contrast to that in Shp2$^{d/d}$ mice, which displayed a pattern resembling the localization during attachment (**Fig 2H,** arrowhead indicated in the below two images). These cellular observations confirmed that the implantation process was delayed in Shp2$^{d/d}$ mice.

Next, we studied the molecular mechanism by which Shp2 affects embryo attachment. Both estrogen and progesterone control the implantation of embryos. Here, ERα and PR were comparably detected in both Shp2$^{f/f}$ and Shp2$^{d/d}$ females, indicating that Shp2 deletion does not influence the protein levels of hormone receptors (**S2E–S2G Fig**). Furthermore, the expression of genes downstream of estrogen and progesterone in mouse uterine tissue was examined. Mucin-1 (Muc1) and lactoferrin (Ltf), proliferating uterine epithelial cell markers [43,44], disappear when the embryo adheres. However, the protein levels of Muc1 and Ltf were still maintained at high levels in Shp2$^{d/d}$ mice, as determined by immunochemistry or immunofluorescence staining (**Fig 2I and 2J**). Assessment of their mRNA levels by RT-PCR confirmed the abnormal upregulation of the *Muc1* and *Ltf* transcripts in Shp2$^{d/d}$ mice (**Fig 2K, p<0.01**). The levels of heart and neural crest derivative-expressed 2 (Hand2), amphiregulin (Areg) and Lif), which are critical regulators of embryo implantation [13,45], were downregulated in Shp2$^{d/d}$ mice (**Fig 2L, p<0.01**).

These results indicate that the uterine stromal protein Shp2 plays a critical role in om-time implantation by regulating the expression of functional genes induced by estrogen and progesterone.

## Shp2 is indispensable for stromal decidual transformation in mice

Invasion of an embryo into the uterine wall stimulates stromal cells to undergo a functional and morphological transformation, a process termed decidualization [4]. We collected uteri from D6-8 of pregnancy and examined the histologies of the implantation sites (**Fig 3A**). Significant decreases in the sizes and weights of the implantation sites (ISs) were observed in mice lacking Shp2 on D6-8 of pregnancy compared with those of Shp2$^{f/f}$ mice (**Fig 3A and 3B**). The embryos implanted into the Shp2$^{d/d}$ uteri were observably smaller on days 7–8 of pregnancy (**Fig 3C**). Protein and transcription expression of decidual tissue prolactin-related protein (Dtprp), a classical marker of decidualization in stromal cells [46], was prominently reduced in the antimesometrial decidual cells of Shp2$^{d/d}$ uteri on day 8 of pregnancy, suggesting a blockade of stromal cell differentiation in these mice (**Fig 3D–3F**). In addition, the expression levels

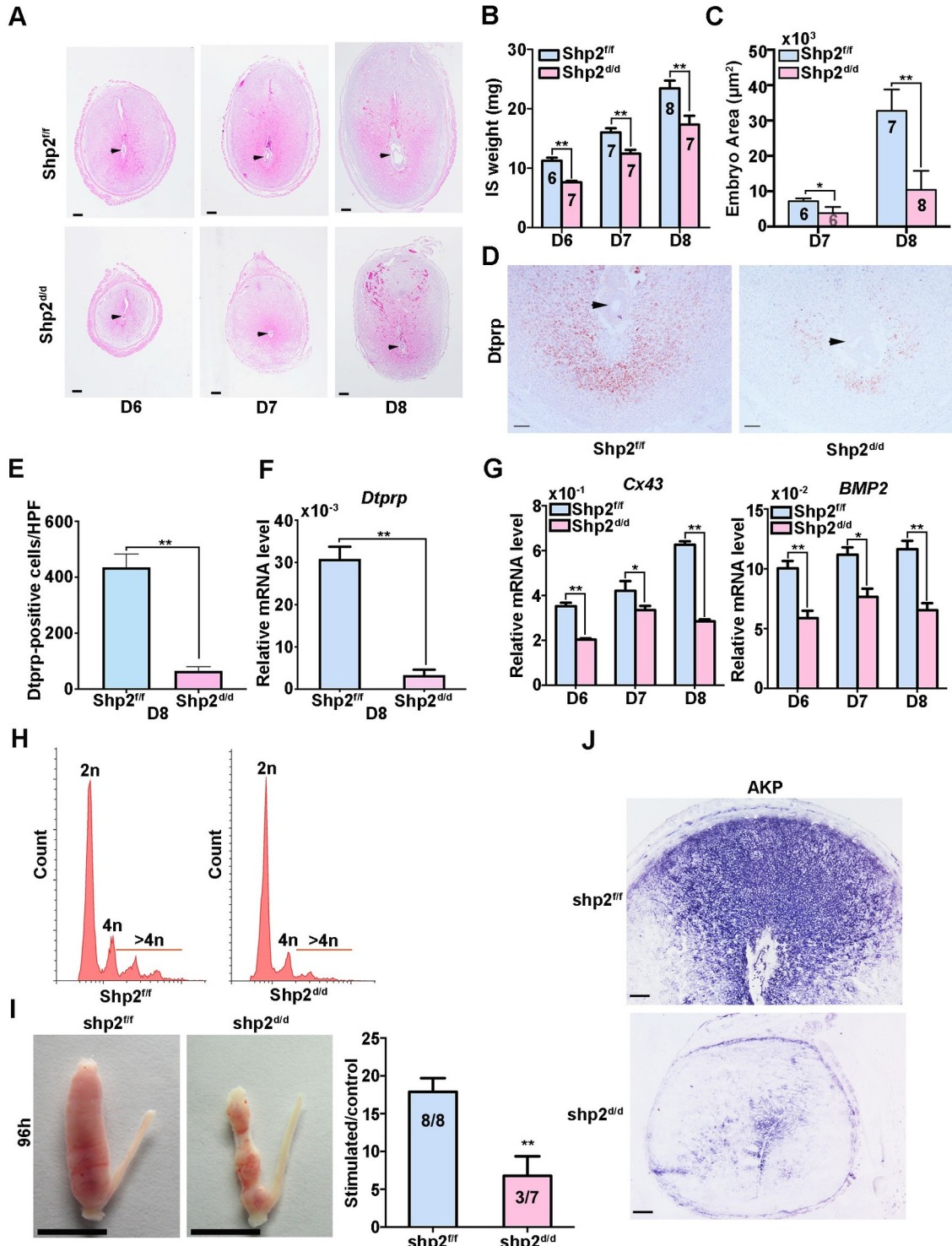

**Fig 3. Shp2 deficiency impairs the process of uterine decidualization, leading to embryo growth retardation and lethality. (A)** ISs collected from Shp2$^{f/f}$ (n = 6/7/8) and Shp2$^{d/d}$ (n = 7/7/7) mice at sequential time points (D6-D8) during pregnancy and stained with hematoxylin and eosin. The arrowheads indicate the embryos. Scale bars, 200 μm. **(B)** Average wet weights of ISs on days 6–8 of pregnancy. **(C)** Average areas of embryos in Shp2$^{f/f}$ (n = 6/7) and Shp2$^{d/d}$ mice (n = 7/8) on D7-D8 of pregnancy. The numbers correspond to the number of pregnant mice examined. **(D)** Dtprp immunostaining of ISs collected from Shp2$^{f/f}$ (n = 6) and Shp2$^{d/d}$ mice (n = 6) on D8. Scale bars, 100 μm. **(E)** Quantitative analysis of Dtprp-positive cells of panel D (n = 6). HPF: high power field. **(F)** Transcription expression of *dtprp* in D8 IS collected from Shp2$^{f/f}$ (n = 6) and Shp2$^{d/d}$ mice (n = 6) as determined by qRT-PCR. **(G)**

Quantitative PCR detection of relative *Bmp2* and *Cx43* expression levels of D6-D8 IS collected from Shp2[f/f] and Shp2[d/d] mice. **(H)** Flow cytometric analysis of cell ploidy of PI-stained stromal cells isolated from day 8 of pregnancy (n = 6). **(I)** The top images depict the gross uterine morphologies of Shp2[f/f] (n = 8) and Shp2[d/d] mice (n = 7) collected at 96 h after the injection of oil into one horn (left) to stimulate stromal cell decidualization. The other horn (right) was untreated and used as a control. Scale bars, 1000 μm. The bottom figure shows the ratio of the wet weights of oil-injected and control horns collected from Shp2[f/f] and Shp2[d/d] mice at 96 h after the application of the decidual stimulus. **(J)** Differentiation in Shp2[f/f] (n = 8) and Shp2[d/d] (n = 7) uteri 96 h after decidual trauma as measured by alkaline phosphatase staining. Scale bars, 100 μm. The data are presented as the mean ± SD from at least 6 mice in each group or three independent experiments. Statistical differences are indicated as follows: *P<0.05, **P<0.01.

of Connexin43 (Cx43) and Bmp2, which are essential for normal decidualization [14,47], were dramatically reduced in Shp2[d/d] females on days 6–8 of pregnancy (**Fig 3G**). These observations suggested that the loss of Shp2 caused defective decidualization of uterine stromal cells.

We next assessed the state of polyploidy, a hallmark of mature decidual cells [4,8,9]. As shown in **Fig 3H**, the analysis of decidual cells by flow cytometry after DNA staining revealed a significant decrease in the proportion of cells that possessed more than 4 copies of genomic DNA (indicated as >4n) in Shp2[d/d] mice. Thus, the defect in decidualization observed in Shp2[d/d] uteri was associated with attenuated terminal maturation and reduced polyploidy.

To eliminate possible effects of deferred embryo implantation on decidual growth restriction in Shp2[d/d] female mice, we induced an artificial decidual reaction [4]. One uterine horn was administered a decidualization stimulus, whereas the contralateral horn served as the unstimulated control. Upon examination of the uteri 96 h after artificial decidual stimulation, the Shp2[f/f] uteri exhibited robust decidual responses (**Fig 3I,** top left image), whereas the decidual responses were notably reduced in stimulated Shp2[d/d] uterine horns (**Fig 3I,** top right image), causing the weight ratios to be reduced by >60% (**Fig 3I,** bottom figure). Moreover, intense expression of alkaline phosphatase, a marker of stromal cell differentiation [4], was observed in oil-induced Shp2[f/f] uteri (**Fig 3J,** top image) but not in Shp2d/d uteri (**Fig 3J,** bottom image).

## Shp2 is necessary for the decidualization of human uterine stromal cells

To further explore the role of Shp2 in decidualization, the decidualization of immortalized hESCs was artificially induced with EPC treatment (estrogen E2, progesterone medroxyprogesterone acetate (MPA), and cAMP) in vitro [48]. Human ESCs transform from fibroblast-like cells in the proliferative phase to epithelial-like cells exhibiting cytoplasmic expansion, large pale nuclei, and rounded shapes in the secretory phase [49]. After 3 days of induction, the hESCs showed obvious morphological changes (**S3A Fig**), and the protein levels of decidual markers [49], insulin-like growth factor-1 (IGFBP1) and prolactin (PRL) were significantly upregulated (**S3B and S3C Fig**), indicating the successful induction of artificial decidualization in vitro. During hESC decidualization, the protein and mRNA levels of SHP2 were gradually increased and reached a remarkably high level in differentiated hESCs (**Fig 4A and 4B**), implying that SHP2 plays an important role in the process of decidualization.

To explore the function of SHP2 in hESCs, its expression in hESCs was transiently knocked down with an siRNA (**S4B and S4C Fig**). As a result, the decidualization morphological change was inhibited (**S4A Fig**), and the expression levels of the decidual markers IGFBP1 and PRL were clearly lower than those in control cells after 72 h of EPC treatment (**S4D Fig**). Furthermore, we constructed SHP2 knockout cell lines using the CRISPR-Cas9 strategy (**Fig 4C**). The knockout cells could not undergo decidualization, as evidenced by their low expression of IGFBP1 and PRL after 72 h of decidualization induction (**Fig 4D**). In addition, SHP2 was transiently overexpressed in hESCs via an adenovirus-mediated method; the cells exhibited a decidual-like morphology, as they were plump and rounded (**S4E Fig**), and the mRNA level of

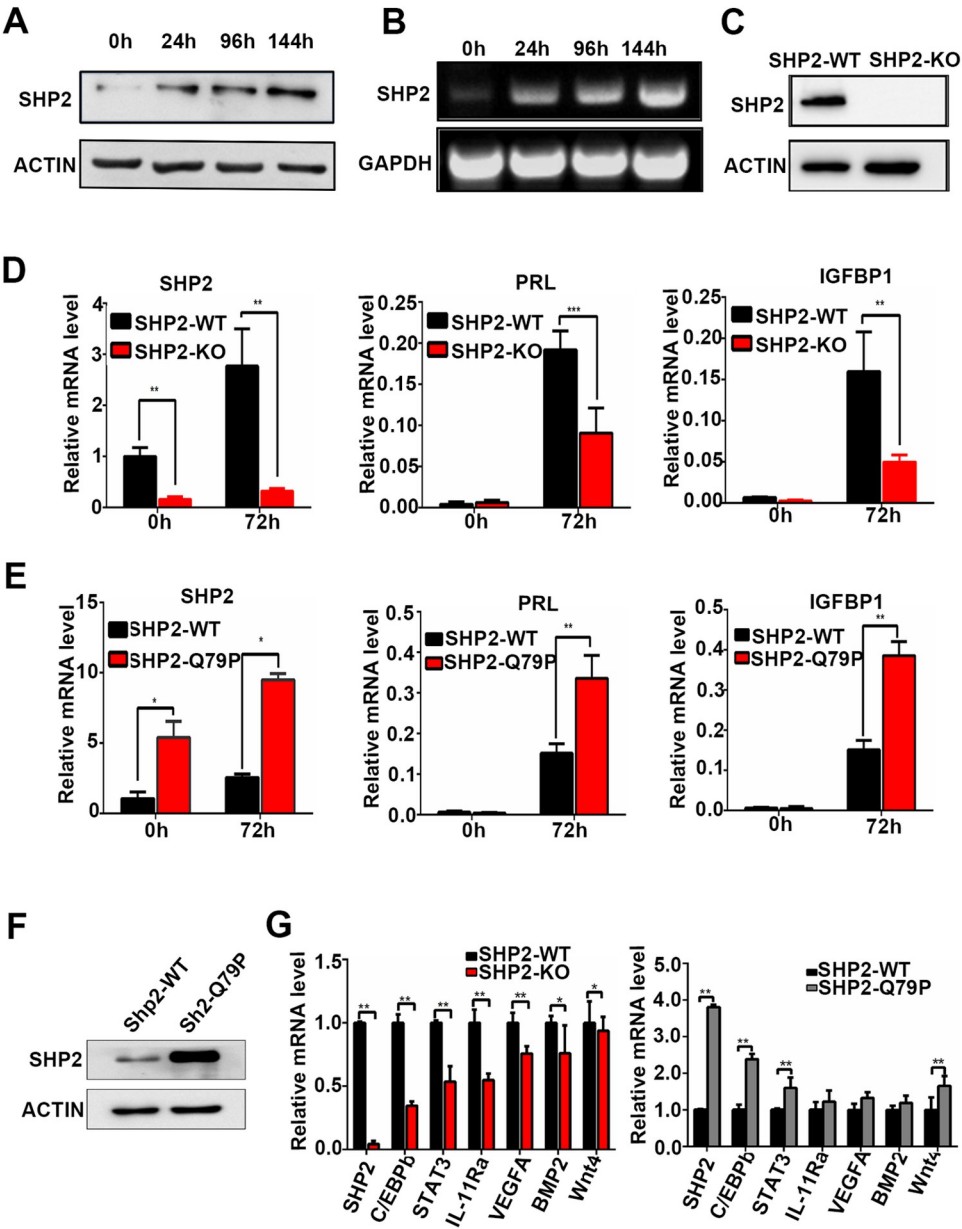

**Fig 4. SHP2 is necessary for the decidualization of human endometrial stromal cells.** (A) Protein level of SHP2 in human immortalized endometrial stromal cells (hESCs) during decidualization as detected by western blotting. (B) SHP2 mRNA levels in decidualized hESCs as determined by real-time fluorescence quantitative PCR. (C) Protein expression of SHP2 in hESCs with stable knockout as determined by western blotting. (D) The mRNA levels of SHP2, IGFBP1 and PRL in decidual hESCs after knockout of SHP2. (E) The mRNA levels of SHP2, IGFBP1 and PRL in decidual hESCs overexpressing SHP2. (F) Protein expression of SHP2 in hESCs stably overexpressing SHP2 as determined by western blotting. (G) The expression of decidual function genes in hESCs with SHP2 knockout (left figure, SHP2-KO) or SHP2 overexpression (right figure, SHP2-Q79R) as determined by real-time fluorescence quantitative PCR. Actin was used as a total protein control in the western blotting assay. GAPDH was used as an internal reference in RT-PCR. The data are presented as the mean ± SD from at least three independent experiments. Statistical differences are indicated as follows: *P <0.05, ** P <0.01.

the decidual marker IGFBP1 was obviously enhanced (S4F Fig). A stable SHP2 overexpression cell line was also constructed by lentivirus infection (Fig 4F), and these cells exhibited an enhanced decidualization ability, as indicated by significant increases in the expression levels

of PRL and IGFBP1 expression (**Fig 4E**), suggesting that the overexpression of SHP2 promotes the decidualization of human stromal cells. The quantitative analysis of protein and mRNA expressions of Shp2 in this figure had been presented in **S5 Fig**.

To portray the mechanism of SHP2 regulation in hESC decidualization, we performed RNA sequencing analysis to screen for genes related to decidualization in hESCs with or without SHP2. Gene expression in hESCs was analyzed at 0, 4, 24, and 72 h of induction, and a total of 5891 differentially expressed genes (DEGs) were identified based on a standard log2 (fold change)| > 0 and a p-adjusted p value (padj) <0.05 (**S1 Data**). Then, the DEGs were clustered and the result was displayed in a Heatmap (**S6 Fig**)**,** in which X axis was the sample cluster, Y axis was the genes cluster, and color change showed the value of log2 (FoldChange). During the induced decidual differentiation, the gene expression profiles were significantly different at each induction time point in WT hESCs, indicating that numerous genes were expressed to response to complex biological functions and cellular changes occurring during decidual differentiation. In SHP2-knockout hESCs, the gene expression profiles were also significantly different between 0h,4h, and 24h of induction. However, there were very similar gene expression profiles between 24 h and 72 h of induction, indicating the decidual process was interrupted in hESCs lacking SHP2 after 24 h of induction (**S6 Fig**). These results provide genomic evidence that SHP2 deficiency blocks hESC decidualization.

Then, DEGs were displayed in a volcano diagram with the value of -log10(padj) and log2 (FoldChange) of each gene. In total, approximately 100, 3025, 3925 and 2006 DEGs were observed at 0h, 4h, 24h, and 72h of induction time point between the control and knockout cells, respectively (**S7 Fig**). We confirmed the expression of more decidual-related genes in hESCs with SHP2 overexpression or knockout after 72 h of induction. The mRNA levels of *C/EBPβ*, *STAT3*, *interleukin-11 receptor α (IL-11Rα)*, *vascular endothelial growth factor A (VEGFA)*, *BMP2* and *WNT4* were remarkably decreased in hESCs lacking SHP2 (**Fig 4G,** left figure), confirming that Shp2 deficiency blocked decidual differentiation. The expression of *C/EBPβ*, *STAT3* and *WNT4* was increased in hESCs overexpressing Shp2 (**Fig 4G,** right figure), consistent with the observation that Shp2 regulates the decidualization process.

## Shp2 mediates cell proliferation to regulate the decidual differentiation of uterine stromal cells

In early decidualization, uterine stromal cells undergo massive proliferation. On D4 of pregnancy, the proliferation of endometrial stromal cells in Shp2[d/d] uteri was attenuated (**Fig 2E and 2F**). Furthermore, the uteri of Shp2[d/d] mice exhibited a decreased number of cells stained positive for phospho-histone H3 (pH3, a marker of the mitosis stage of the cell cycle) on day 6 of pregnancy (**Fig 5A**). We isolated stromal cells from uteri on day 4 of pregnancy (**S8 Fig**). In response to 10 nM E2 and 1 μM P4 treatment, primary stromal cells exhibited minimal proliferation, and the pH3-positive cell numbers were dramatically reduced upon deletion of Shp2 (**Fig 5B**). Moreover, the percentage of Shp2[d/d] stromal cells in the G2/M phase was significantly decreased (**Fig 5C**). The cyclin B1 and Cdk1 expression levels were markedly reduced in Shp2-null uterine stromal cells after 24 h of decidual stimulation. These observations confirmed a mitosis deficiency in Shp2[d/d] stromal cells (**Fig 5D**).

The expression of KI67, a cell proliferation marker, was clearly lower in SHP2-knockout hESCs than in control cells (Shp2-WT), indicating that the growth of hESCs was also suppressed by the deletion of Shp2 (**Fig 5E**). During the induction of hESC decidualization by EPC treatment, G2/M checkpoint genes were significantly enriched by SHP2 loss as determined by gene set enrichment analysis (GSEA) (**Fig 5F**), indicating that the cell cycle process was potentially disrupted by Shp2 deficiency. As a confirmation, the number of Shp2-deficient

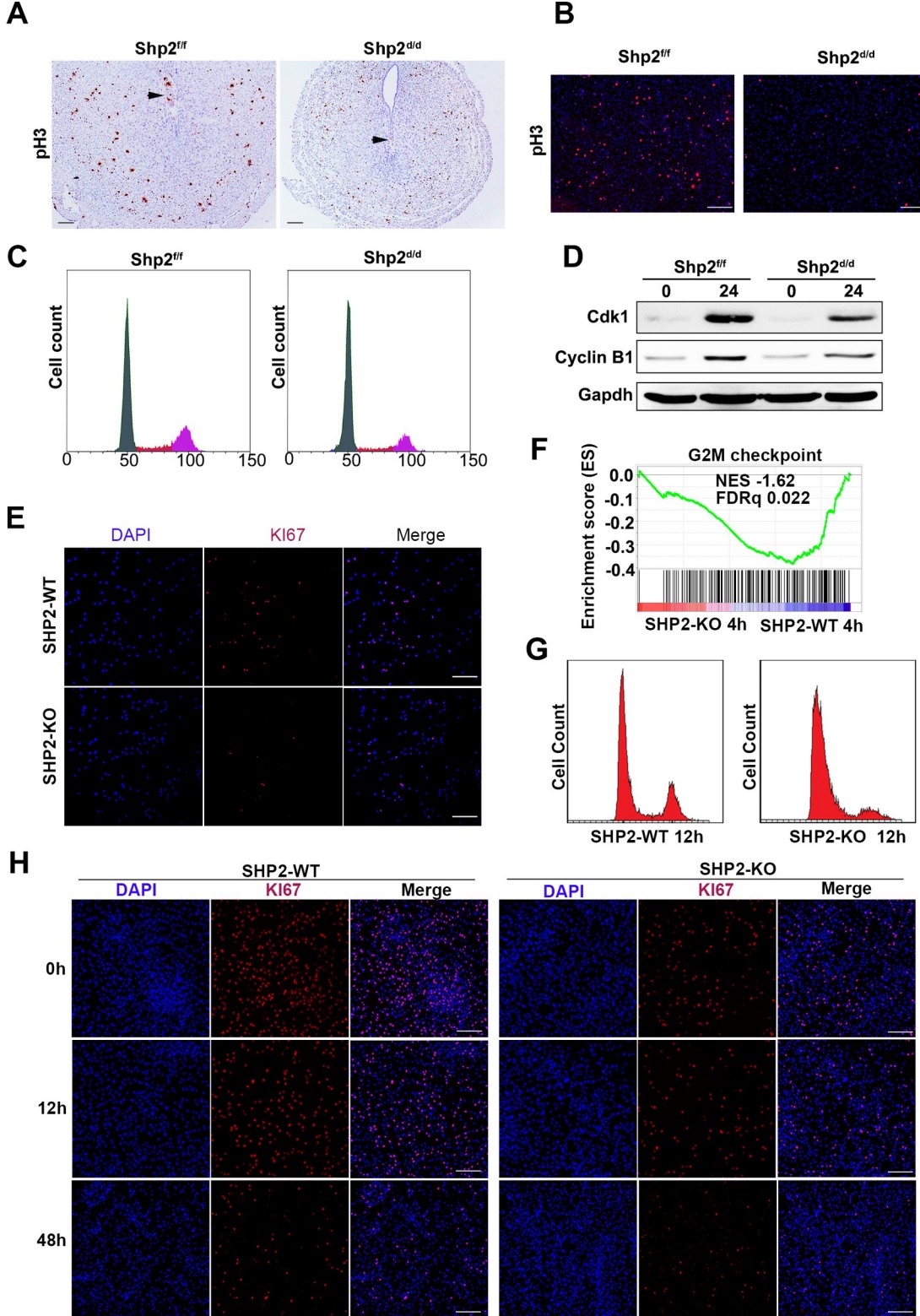

**Fig 5. Shp2 knockout disrupts cell proliferation to prevent the differentiation of stromal cells. (A)** Immunohistochemical detection of phospho-histone H3 (pH3) at the ISs of Shp2$^{f/f}$ (n = 6) and Shp2$^{d/d}$ mice (n = 6) on D6. Scale bars, 100 μm. **(B)** Immunocytochemical detection of pH3 in Shp2$^{f/f}$ (n = 6) and Shp2$^{d/d}$ (n = 6) primary stromal cells treated with the decidual regimen for 24 h. Scale bars, 100 μm. **(C)** Cell cycle distribution of PI-stained hESCs at different time points during the decidual

induction of mouse primary stromal cells as determined by flow cytometry (n = 6). The percentages of cells in the G0/G1, S and G2/M phases as determined by flow cytometry are shown in the right figures. **(D)** CyclinB1 and Cdk1 in mouse primary stromal cells with or without decidual treatment for 24 h. Gapdh served as the loading control. **(E)** The proliferation of wild-type and SHP2-KO hESCs was evaluated by IF staining with a KI67 antibody. Scale bars, 100 μm. **(F)** GSEA plot evaluating the changes in the indicated gene signatures in response to the G2M checkpoint (n = 189) in SHP2-KO hESCs compared with SHP2 WT hESCs after 24 h of decidual induction. NES, normalized enrichment score. **(G)** The cell cycle distribution in PI-stained stromal cells treated with the decidual regimen for 12 h was analyzed by flow cytometry. **(H)** KI67 staining in hESCs during the decidual process. Scale bars, 100 μm.

cells in the G2/M stage remained low throughout the process (**Fig 5G,** right image), while the number of WT hESCs in the G2/M stage obviously peaked after 12 h of EPC treatment (**Fig 5G,** left image). In addition, staining of KI67 also revealed strong proliferation in the early stage of hESC decidualization, while the proliferation of hESCs lacking SHP2 remained low (**Fig 5H**).

These results indicate that Shp2 knockout hinders the cell cycle and causes decidual differentiation failure.

## Shp2 regulates the decidual response of stromal cells by participating in a variety of signaling pathways

Subsequently, we performed Kyoto Encyclopedia of Genes and Genomes (KEGG) analysis of the DEGs at each time point to elucidate the signaling pathways affected by Shp2. The impact of Shp2 deletion was obvious after 4 h of hormone and factor treatment. DEGs were enriched in the PI3K/AKT, MAPK, cAMP, transforming growth factor β (TGF-β), HIPPO, and JAK-STAT pathways. At 24 h of induction, these signals were still affected by the loss of SHP2 (**S9 Fig**).

To confirm these findings based on RNA sequencing, we explored the effects of Shp2 on the signaling pathways involved in stromal cell decidualization. In mice, the expression of phospho-ERK1/2 was prominent in uterine stromal cells in Shp2$^{f/f}$ mice on days 6–7 of pregnancy, whereas deletion of Shp2 drastically abolished ERK1/2 activation, as determined by immunohistochemistry (**Figs 6A and S10A**) and western blot (**Figs 6B and S10B**). In mouse primary stromal cells treated with 10 nM E2 and 1 μM P4, ERK1/2 activation was substantially attenuated by Shp2 deletion (**Fig 6C,** right band in the top panel). The phosphorylation of CCAAT/enhancer binding protein β (C/EBPβ), a known substrate of ERK1/2, was increased in Shp2$^{f/f}$ mice (**Fig 6C,** second band in the top fourth panel) but reduced in stromal cells lacking Shp2 (**Fig 6C,** fourth band in the top fourth panel), while the total C/EBPβ levels were not altered (**Fig 6C,** the top fifth panel). ERK was phosphorylated in hESCs after 30 min of EPC induction treatment (**Fig 6D,** second band in the top panel), but EPC stimulation failed to activate ERK in the absence of Shp2 (**Fig 6D,** fourth band in the top panel). The quantitative analysis of protein level had been showed in **S10C and S10D Fig**. These results suggest that knockout of Shp2 in uterine stromal cells inhibits ERK signal transduction.

Contrary to the ERK signal, AKT and its downstream protein FOXO1 were highly activated in untreated hESCs (**Figs 6E and S10E,** the first band in the top first and third panel). When hESCs were exogenously stimulated by EPC induction for 1 h, AKT and FOXO1 were inactivated in hESC (**Figs 6E and S10E,** second band in the top first and third panels). But, p-AKT and p-FOXO1 were still kept in high level in hESC without Shp2 after EPC induction, which was contrary to the observation in wild type hESC (**Figs 6E and S10E,** the fourth band in the top first and third panel). These results indicate that Shp2 is involved in the negative regulation of decidual induction on the AKT-pFOXO1 signaling pathway.

The transcription factor STAT3 is activated by several signals to mediate the decidualization of uterine stromal cells and is reportedly regulated by Shp2. GSEA also revealed that

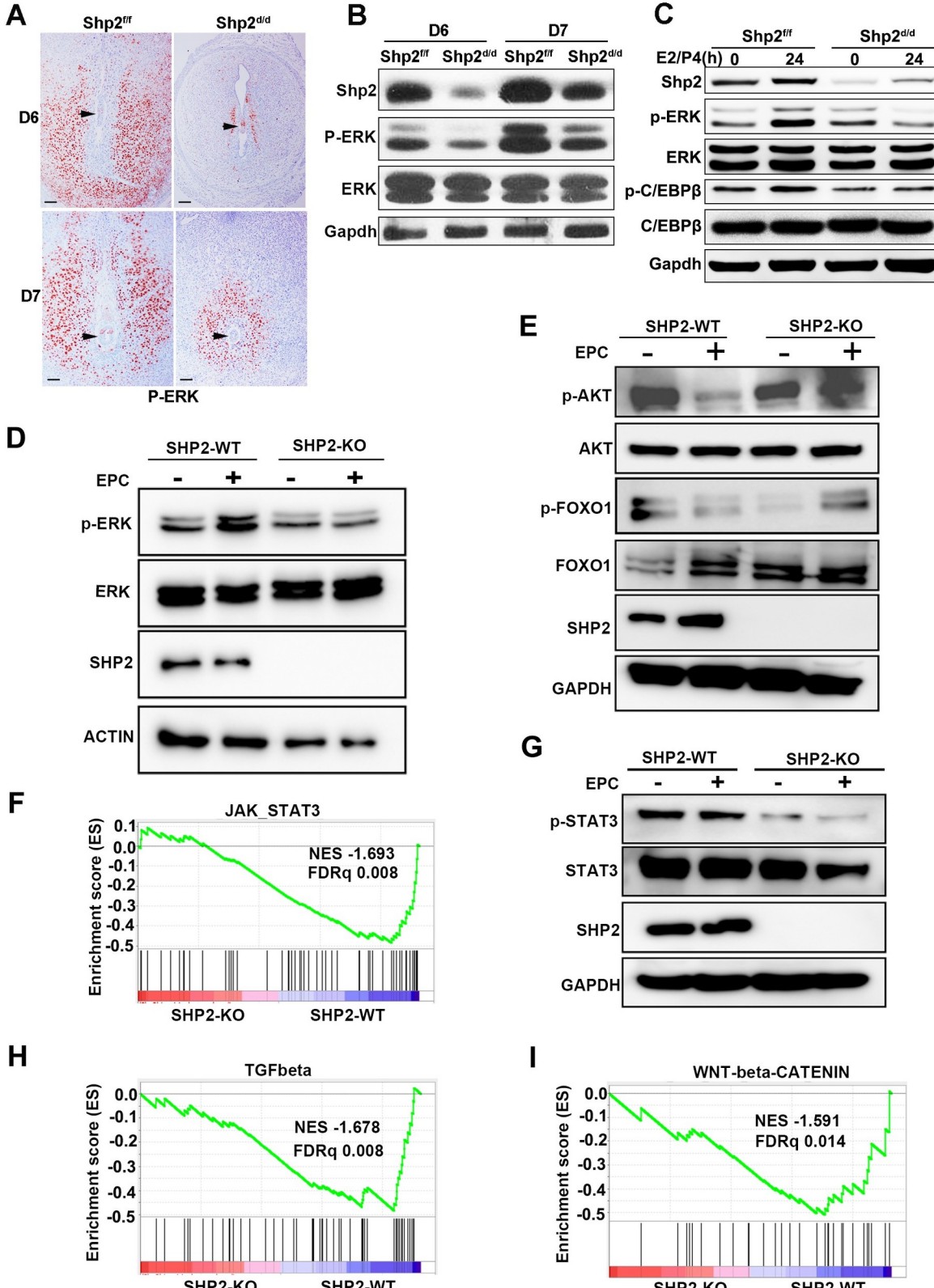

**Fig 6. Shp2 is involved in the regulation of multiple ERK, AKT, and STAT3 signaling pathways in immortalized hESCs. (A)** Immunohistochemical detection of phospho-ERK1/2 in the ISs of Shp2$^{f/f}$ (n = 6) and Shp2$^{d/d}$ mice (n = 6) on D6 and D7. The black arrowheads indicate the embryos. **(B)** The protein levels of Shp2, phospho-ERK1/2 and total ERK1/2 in the ISs of Shp2$^{f/f}$ and Shp2$^{d/d}$

mice on D6 and D7 as determined by western blotting. Gapdh served as the loading control. **(C)** Assay of Shp2, p-Erk1/2, ERK1/2, p-C/Ebpβ, and C/Ebpβ in mouse primary stromal cells treated with the decidual regimen for 24 h. **(D)** Total protein and phosphorylated protein levels of ERK in decidally induced hESCs with or without SHP2 as determined by western blotting. Actin served as the internal control. **(E)** Analysis of AKT and FOXO1 activation (p-AKT and p-FOXO1) in decidual hESCs by western blotting. GAPDH was used as the loading control. **(F)** GSEA plot indicating the gene signatures in response to JAK_STAT3 (n = 52) in SHP2-KO hESCs compared with SHP2-WT hESCs after 24 h of decidual induction. **(G)** Phosphorylation of STAT3 in hESCs during the decidual process. **(H&I)** Enrichment of the TGF-β (n = 51) and WNT/β-catenin (n = 30) signaling pathway genes in SHP2-KO hESCs compared with SHP2-WT hESCs after 24 h of decidual induction as determined by GSEA. NES, normalized enrichment score. EPC: E2, MPA and cAMP.

JAK-STAT3 signaling pathway genes were significantly enriched by SHP2 deletion (**Fig 6F**). We next evaluated the phosphorylation level of STAT3 in hESCs and found that the expression levels of both basal and induced p-STAT3 were reduced by Shp2 deficiency after one hour of EPC treatment (**Figs 6G and S10F**, fourth band in the first and third panels). Interestingly, TGF-β (BMP) and Wnt/β-catenin signaling pathway genes, including *BMP2*, *SMAD1*, *FKBP1A*, *SMURF2*, *ID1*, *TGFBR1*, *WNT5B*, *CTNNB1*, *FZD1*, *AXIN2*, *FRAT1*, *HDAC2*, *NCSTN*, *RBPJ*, and *DVL2*, were enriched by SHP2 deletion as determined by GSEA (**Fig 6H & 6I**), indicating that SHP2 is associated with these two key pathways regulating decidual signaling.

These results suggest that SHP2 regulates the responses of stromal cells by participating in a variety of signaling pathways (**S11 Fig**).

## Discussion

In this study, we demonstrated that the uterine stromal protein Shp2, a key factor that mediates and integrates multiple signals, is essential for "on-time" embryo implantation and complete decidualization.

### Shp2-mediated signals in epithelial and stromal cells together support "on-time" embryo attachment

Embryo implantation proceeds from the initial attachment of the embryo to the uterine luminal epithelial surface. Uterine epithelial-mesenchymal dialog dominated by estrogen and progesterone is essential for embryo implantation [50]. Here, Amhr2-Cre-mediated Shp2 deficiency deferred embryo implantation. Previously, we demonstrated that Shp2 ablation in both epithelial and stromal cells completely blocked embryo implantation [38]. Although the knockout efficiency of Amhr2-Cre might affect the phenotype of the mouse model [51,52], the discrepancy between the two models still indicated that Shp2-mediated signals in epithelial cells supported, at least in part, embryo attachment. Indeed, Shp2 has been demonstrated to interact with the ER to mediate the expression of genes downstream of estrogen, including PR, in uterine epithelial cells [38]. The deletion of COUP-TFII in Amhr2-Cre and PR-Cre mouse models also resulted in different phenotypes [53,54]. In summary, this observation provides new evidence of an association between the uterine epithelium and stroma and indicates that the Shp2-mediated signals in epithelial and stromal cells together support embryo attachment.

### Shp2-mediated signaling in stromal cells is essential for endometrial epithelial remodeling

The establishment of an endometrial receptive state involves drastic changes in the morphology and function of endometrial cells, including the remodeling of epithelial cells and the proliferation and differentiation of stromal cells [2,3,7,8]. In mice, uterine epithelial cell proliferation is stopped before embryo attachment. During embryo attachment, luminal

epithelial cells disappear, and stromal cell proliferation increases at the implantation site [8,9]. Here, Shp2 deficiency in stromal cells disrupted the cessation of cell proliferation and the disappearance of epithelial cells, suggesting that Shp2 governs the establishment of an endometrial receptive state by regulating endometrial epithelial remodeling.

The establishment of endometrial receptive states involves complex signals, including estrogen and progesterone, as well as multiple local growth factors. Preovulation, estrogen stimulates stromal cells to secrete paracrine factors, such as IGF1 and FGF, to thereby regulate epithelial proliferation [9]. After the corpus luteum is formed, progesterone induces the expression of Hand2 in stromal cells, thereby inhibiting the production of the proliferation signals FGF and IGF1 [16,55,56]. Shp2 has been demonstrated to interact with the ER in uterine epithelial cells [38], breast epithelial cells [57] and adipocytes [58] and to regulate cell proliferation by mediating FGF and IGF1 signaling [9]. The expression of several PR- and ER-target genes (Muc1, Ltf, Hand2, Areg and Lif) was shown to be abnormal in the uterine tissues of Shp2 knockout mice. However, deletion of Shp2 did not affect the expression of ER or PR. These results suggest that Shp2 is a signaling protein downstream of estrogen and progesterone receptors that mediates the biological regulation of hormones to facilitate on-time implantation.

## Shp2-mediated signals in stromal cells are essential for stromal decidualization

In mice, the decidual reaction first spreads throughout the antimesometrial region and then to the mesometrial side (the presumed site of placentation) [4]. Polyploidy with large mono- or binuclei is a characteristic of terminally differentiated decidual cells at D8 [4]. However, in mouse uterine stromal cells lacking Shp2, the progress of decidualization was seriously disturbed, and polyploidy numbers were low. Artificial decidualization induction in mice dramatically increases the size of the uterus and results in the enlargement and differentiation of the stromal cell layer, similar to embryo-induced decidualization during normal pregnancy [4]. The suppressive effects of stromal Shp2 deficiency on decidualization were also confirmed using an artificially inducted decidualization model in vitro. Humans undergo spontaneous decidualization driven by increasing progesterone levels postovulation and increasing local cAMP production during the middle-to-late secretory phase of each menstrual cycle [8,48]. Here, the decidual differentiation of hESCs was induced by progesterone, estrogen and cAMP in vitro, and Shp2 was demonstrated to be necessary for this process.

Shp2 exhibits spatiotemporal expression during the decidualization process in both human and mouse uterine stromal cells and high expression levels in terminal decidual stromal cells, indicating that it may play a key role in the function of decidual stromal cells. However, the decidualization of stromal cells was blocked in our study. Thus, the effects of Shp2 on the function of terminal decidual stromal cells need to be explored.

## Shp2 mediates cell proliferation and thereby regulates the decidual differentiation of stromal cells

During decidualization, uterine stromal cells undergo massive proliferation, followed by differentiation into distinct decidual cells. In mice, stromal cells surrounding the implanted blastocyst undergo proliferation to form the primary decidual zone (PDZ) between day 5 and day 6, and stromal cells adjacent to the PDZ then continue to proliferate and differentiate into decidual cells to form the secondary decidual zone (SDZ) at the antimesometrial pole on day 8 [2–4]. However, the proliferation of uterine stromal cells is suppressed in Shp2 knockout mice, followed by retarded and defective stromal decidualization. In some studies as well as in our

investigation, hESC mitosis was observed in the early stage of decidualization, as shown in Fig 5H [22,59]. However, the number of hESC mitotic cells was low during decidualization. Shp2 shows an important ability to regulate cell proliferation through the ERK signaling pathway triggered by EGF, IGF-1, insulin, and other factors [25]. The activation of ERK and its downstream proteins C/EBPβ, cyclinB1 and CDK1 is seriously suppressed in uterine stromal cells. Thus, the Shp2-ERK signaling pathway may be an important cell cycle regulator that controls the G2-to-M transition in stromal cells.

## Shp2 mediates and integrates multiple signaling pathways in stromal cells during decidualization

During decidualization, stromal cells undergo proliferation, proliferation cessation, cytoskeletal reorganization, cell adhesion, and extracellular matrix changes to adapt to the decidual phenotype [4,22]. Progesterone and estrogen have an initial and rapid decidualization effect, although whether estrogen is required to induce the decidualization of human stromal cells in vitro is controversial [17,22,59]. The interactions of cytoplasmic signaling pathways mediate and exert the synergistic effects of hormones and local factors. Various cytoplasmic signaling pathways are involved in the regulation of cAMP, progesterone and estrogen [17,22,59]. In addition to the regulatory effect of Erk-C/EBPβ1 on cell proliferation, PI3K/AKT and JAK--STAT regulate morphological and functional differentiation by mediating the signal transduction of cAMP, progesterone, EGF and Ihh [20,21]. During the decidualization of uterine stromal cells, cAMP activates PLD1 to promote the binding of AKT and PP2A [60], thereby inactivating AKT. This phenomenon is followed by dephosphorylated FOXO1 entering the nucleus and binding to PGR, STAT5, and other molecules to initiate decidualization [21,61]. C/EBPβ mediates P4 signaling to regulate the expression of IGFBP1 and directly targets STAT3 to activate IL-11 signaling and thereby accelerate decidualization [62,63]. STAT3 can regulate the PGR target genes HOXO10/11 in response to $P_4$ signals and thereby mediate decidual differentiation [64], and FoxM1 can also target the binding of STAT3 to enhance cell differentiation [65]. Here, SHP2 deficiency induced the activation of AKT and FOXO1, resulting in the failure of FOXO to enter the nucleus. BMP2-WNT4 signals are activated by progesterone and play an important role in the decidual process [15,66]. The absence of BMP2 in the uterus leads to decidual defects in mice [14]. Exogenous BMP2 can induce the differentiation of mouse and human uterine stromal cells [14,66]. During hESC decidualization, BMP (TGF-β) and WNT/β-catenin signaling genes were enriched by the loss of SHP2 as determined by GSEA. All of the above mentioned signaling pathways were found to interact with Shp2 in previous studies [25]. Therefore, these results indicate that Shp2 regulates multiple signaling pathways to exert different functions at different stages of decidualization (**S11 Fig**).

## Conclusions

In summary, Shp2 is expressed in the peri-implantation uterus in a spatiotemporal manner and exerts different functions at different stages of embryo implantation. Consistent with its diversified regulation, Shp2 mediates and integrates different signaling pathways associated with embryo attachment, uterine receptivity and stromal decidualization. Therefore, Shp2, as an important node of the signaling network, is important for the treatment of diseases related to fertility. Our findings not only importantly advance our understanding of this cell type-specific function of Shp2 in peri-implantation biology but may also provide a mechanistic foundation to more effectively diagnose and/or treat women with recurrent implantation failure or early pregnancy loss.

## Materials and methods

### Ethics statement

All animal experiments were performed according to guidelines approved by the Animal Welfare Committee of Research Organization (X201611) of Xiamen University (Xiamen, China).

### Animals and treatment

Shp2$^{flox/fox}$ mice were crossed with anti-Müllerian hormone receptor II-Cre (Amhr2-Cre) mice (purchased from MMRRC, USA) to generate conditional-knockout animals (Shp2$^{d/d}$). Amhr2 is the anti-Müllerian hormone (AMH) type II receptor (also known as Müllerian inhibiting substance type II receptor), which is expressed in the mesenchyme of fetal Müllerian ducts, the anlagen of the uterus, oviducts, cervix and upper portion of the vagina. Shp2$^{f/f}$ female littermates served as controls. All animals used in the experiments were housed at the Xiamen University Laboratory Animal Center, had free access to regular food and water and were cared for in accordance with the institutional guidelines for the care and use of laboratory animals.

To induce artificial decidualization, adult mice were ovariectomized. After a 2-week rest period, the mice were treated with subcutaneous injections of E2 (100 ng, Sigma) for 3 consecutive days and then allowed to rest for 2 days. The mice were then administered 6.7 ng of E2 together with 1 mg of P4 (Sigma) via subcutaneous injection daily for the remaining days. After the third injection of P4 together with E2, 25 μl of sesame oil (Sigma) was injected into the lumen of one uterine horn to induce decidualization, with the contralateral horn serving as a control.

To study ovarian responses to exogenous gonadotropins, 21-day-old immature female mice were intraperitoneally injected with 5 IU pregnant mare serum gonadotropin (PMSG, Sansheng, Ningbo, China) to stimulate preovulatory follicle development, followed by 5 IU human chorionic gonadotropin (hCG, Sansheng, Ningbo, China) 48 h later to stimulate ovulation and luteinization.

A 6-month breeding experiment was conducted to evaluate the effect of Shp2 ablation on female fertility. Briefly, Shp2$^{f/f}$ and Shp2$^{d/d}$ females were placed in a cage at a 1:1 ratio with WT males of proven fertility for 6 consecutive months. During the six-month pregnancy assay, the mice were observed every day after pregnancy (significant signs of baby bulge), and the number of pups per litter was recorded. The pups were separated from their mothers after being weaned. At the end of the experiment, the total number of litters was calculated.

### Tissue collection

Adult female mice were mated with fertile WT males to induce pregnancy. The day the vaginal plug appeared was considered day 1 of pregnancy. Day 4 of pregnancy was validated by recovering embryos from the uteri. The implantation sites on D5 and 6 were visualized by intravenously injecting 0.1 ml of 1% Chicago blue dye (Sigma). Mice were euthanized by cervical dislocation at the designated time points. Upon dissection, uterine tissues were collected and either fixed in 4% paraformaldehyde for histology or frozen at -80˚C for RNA/protein extraction.

### Primary culture of mouse uterine stromal cells

Uterine stromal cells were isolated and cultured as previously described with minor modifications [67]. Briefly, uterine horns from mice on day 4 of pregnancy were cut into small pieces. The tissues were then placed in Hank's balanced salt solution (HBSS) containing 6 mg/ml dispase and 25 mg/ml trypsin for 1 h at 4˚C, followed by incubation for 1 h at room temperature

and then for 10 min at 37˚C. After discarding the endometrial epithelial clumps, the remaining tissues were incubated again in HBSS containing 0.5 mg/ml collagenase at 37˚C for 30 min. The digested cells were passed through a 70-μm filter to acquire stromal cells. The cells were plated at a density of 5 x $10^5$ cells per 60-mm dish and cultured in phenol red-free Dulbecco's modified Eagle's medium and Ham F-12 nutrient mixture (1:1; DMEM/F12; Gibco) containing 10% charcoal-stripped fetal bovine serum (C-FBS; Biological Industries) and antibiotics. After 2 h of incubation, the medium was replaced with phenol red-free culture medium (DMEM/F-12, 1:1) containing 1% charcoal-stripped FBS, 10 nM E2 and 1 μM P4 to induce decidualization in vitro.

## Culture, treatment and decidual induction of human endometrial stromal cells

Immortalized hESCs were cultured in phenol red-free F12 medium containing 10% carbon-adsorbed serum (D-FBS). To transiently knockdown Shp2, cells were transfected with an siRNA targeting the Shp2 sequence. An hESC cell line with stable Shp2 knockout was engineered by CRISPR/Cas9 technology according to previous reports [38]. Adenoviruses overexpressing Shp2 were added to Shp2-deficient cells and normal cells to restore or enhance the level of Shp2 as previously described [68].

To induce decidual differentiation, hESCs were plated in a 100 cm Petri dish and grew to 90% confluence overnight. After the growth medium was discarded and the cells were washed three times with phosphate-buffered saline (PBS), phenol red-free medium containing 1% D-FBS serum and EPC buffer (10 nM 17β-E2, 1 μM MPA, 0.5 mM cAMP) was added to induce decidualization. After three days, stromal cell decidualization was evaluated by cell morphological analysis and assessment of decidual marker gene expression.

## Immunohistochemistry staining

Uterine samples were fixed in 4% paraformaldehyde and then embedded in paraffin. Tissue sections (5 μm) were deparaffinized, rehydrated, and incubated overnight at 4˚C with primary antibodies. After washing with PBS, the sections were incubated with a horseradish peroxidase-conjugated secondary antibody. Immunoreactivity was detected using a DAB substrate kit according to the manufacturer's protocol (Zhongshan Golden Bridge Biotechnology Co., Beijing). To detect alkaline phosphatase activity, frozen tissue sections (10 μm) were used. The slides were fixed in cold acetone for 15 min and rinsed three times with PBS. The BCIP/NBT kit was used for staining according to the manufacturer's protocol (Zhongshan Golden Bridge Biotechnology Co., Beijing). Information about the antibodies utilized is provided in **S1 Table.**

## Immunofluorescence staining

Stromal cells (mouse or human uterine) were fixed and permeated. Nonspecific binding was blocked with PBS containing 5% bovine serum albumin (BSA, Sigma). The cells were then sequentially incubated with primary antibodies overnight at 4˚C and with a fluorescein isothiocyanate-labeled secondary antibody (Zhongshan Golden Bridge Biotechnology) for 1 h at room temperature. The sections were observed under a Leica fluorescence microscope. Information about the antibodies utilized is provided in **S1 Table**.

## Western blotting

Protein was extracted using RIPA buffer supplemented with protease and phosphatase inhibitors (Roche). The protein concentration was determined with a BCA kit according to the

manufacturer's instructions. Samples containing 40 μg of protein were subjected to sodium dodecyl sulfate-10% polyacrylamide gel electrophoresis and then transferred onto polyvinylidene difluoride membranes. The membranes were blocked with 5% nonfat milk for 1 h at room temperature and incubated with primary antibodies at 4˚C overnight. Immunoreactivity was visualized by incubation with horseradish peroxidase-linked secondary antibodies and treatment with enhanced chemiluminescence reagents. Information about the antibodies utilized is provided in **S1 Table**.

### RNA isolation, quantitative real-time PCR

Total RNA was extracted from mouse uterine tissues and cultured human or mouse uterine stromal cells using TRIzol reagent (Roche). One microgram of RNA was reverse transcribed into cDNA. The mRNA expression levels were measured by SYBR Green (Roche) using the ABI 7500 sequence detector system according to the manufacturer's instructions (Applied Biosystems). The corresponding primer sequences used for real-time PCR were listed in **S2 Table**. All real-time PCR experiments were repeated at least three times.

### RNA sequencing

Total RNA was isolated from cells using an RNeasy Mini Kit (Qiagen) according to the manufacturer's protocol. DNase I in-column digestion was performed to remove genomic DNA. In total, 1 μg of each RNA sample was used as input material for the RNA sequencing preparations. Sequencing libraries were generated using the NEBNext RNA Library Prep Kit for Illumina (NEB, USA) according to the manufacturer's recommendations, and index codes were added to attribute sequences to each sample. The libraries were sequenced on an Illumina NovaSeq platform, and 150 bp paired-end reads were generated. Feature Counts v1.5.0-p3 was used to count the number of reads mapped to each gene. Then, the fragments per kilobase of transcript per million fragments sequenced (FPKM) value of each gene was calculated based on the length of the gene and the number of reads mapped to the gene. Differential expression analysis of two conditions/groups (two biological replicates per condition) was performed using the DESeq2 package (1.16.1). Genes with an adjusted P-value <0.05 as determined by DESeq2 were deemed to be differentially expressed.

### Flow cytometry

Stromal cells (mouse or human uterine) were digested and harvested. After centrifugation, the cell pellet was suspended in PBS and fixed in 70% cold ethanol overnight at 4˚C. After centrifugation, cell sediments were suspended in an appropriate volume of PBS containing 30 mg/ml propidium iodide (PI, Sigma) and 0.3 mg/ml DNase-free RNase A (Sigma). The samples were incubated for 30 min at 37˚C in the dark. After filtration through nylon mesh to remove the cell clumps, the cell suspensions were immediately analyzed by flow cytometry by calculating the proportions of cells in the G0/G1, S and G2/M phases based on DNA histogram data. Each experiment was repeated three times.

### Statistical analysis

Data are presented as the mean ± standard deviation from at least 6 mice in each group or three independent experiments. Differences between two groups of normally distributed data were compared with a t-test using GraphPad Prism 5.0 software. In all cases, p<0.05 indicated statistical significance.

## Supporting information

**S1 Fig. Shp2 is expressed in the peri-implantation mouse uterus in a spatiotemporal manner. (A)** Reverse transcription PCR analysis of wild-type uterine Shp2 mRNA levels on D4-D8 of pregnancy (n = 6). **(B)** Western blot detection of Shp2 protein levels in wild-type uteri from D4-D8. **(C)** Immunohistochemical staining of Shp2 in local area (up) and full tissues (down) of wild-type D4-D7 uteri (n = 6). Scale bars for the top 4 images in panel C, 20 μm; Scale bars for the bottom four images in panel C, 200 μm. The data are presented as the mean ± SD from at least 6 mice in each group or three independent experiments. Statistical differences are indicated as follows: $^*P<0.05$, $^{**}P<0.01$.
(TIF)

**S2 Fig. Ovarian functions remain unaffected in Shp2$^{d/d}$ mice. (A)** Hematoxylin and eosin staining showed similar ovarian histology between Shp2$^{f/f}$ (n = 7) and Shp2$^{d/d}$ (n = 8). Scale bars, 100μm. **(B)** In superovulation assay, Shp2$^{d/d}$ (n = 8) mice ovulated comparable oocytes to that of the control mice (n = 7) at 16 h after hCG. **(C-D)** Serum levels of E2 and P4 in Shp2$^{f/f}$ (n = 7) and Shp2$^{d/d}$ mice (n = 6) on D4 of pregnancy. Number within the bar indicated the number of mice tested. **(E-F)** Immunohistochemistry staining of ER (E) and PR (F) in Shp2$^{f/f}$ (n = 7) and Shp2$^{d/d}$ (n = 6) uteri on day 4 of pregnancy. **(G)** Western blotting of ER and PR in Shp2$^{f/f}$ and Shp2$^{d/d}$ uteri on day 4 of pregnancy. Gapdh serves as loading control. The data are presented as the mean ± SD from at least 6 mice in each group or three independent experiments. Statistical differences are indicated as follows: $^*P<0.05$, $^{**}P<0.01$.
(TIF)

**S3 Fig. Model of human uterine stromal cells induced decidualization in vitro. (A)** The left picture shows the bright field picture of the undifferentiated uterine stromal cells, and the right picture shows the decidual cells induced for 6 days, Scale bar: 30μm **(B)** Real-time fluorescence quantitative PCR technology was used to detect the mRNA expression of decidual marker molecules IGFBP1 and PRL, GAPDH was used as internal reference. **(C)** RT-PCR technology was used to detect RNA expression of IGFBP1 and PRL, GAPDH was used as internal reference. The data are presented as the mean ± SD from at least three independent experiments. Statistical differences are indicated as follows: $^*P <0.05$, $^{**} P <0.01$.
(TIF)

**S4 Fig. The effects of transient knockdown or overexpression of Shp2 on the decidualization of human uterine stromal cells. (A)** Brightfield image of hESCs cultured with decidual inducer for 96h. Con-si: siRNA control, Shp2-Si: siRNA of Shp2, Scale bar: 30μm. **(B)** Shp2 protein in hESC transiently transfected with siRNA of Shp2 or control checked by western blotting assay. GAPDH as internal control. **(C-D)** The mRNA of Shp2, IGFBP1and PRL in knockdown Shp2 hESCs measured by RT-PCR. **(E)** Brightfield image of hESCs transiently overexpressed Shp2. Con-ad: control virus, Q79R-ad: virus expressing Shp2(Q79R), Scale bar: 30μm. **(F)** The top panel shows the protein expression of Shp2 in hESCs transiently overexpressed Shp2 detected by western blotting, GAPDH is the internal reference; The bottom panel shows the mRNA level of IGFBP1 in hESCs transiently overexpressed Shp2 checked by RT-PCR. GAPDH is the internal reference. The data are presented as the mean ± SD from three independent experiments. Statistical differences are indicated as follows: $^* P <0.5$, $^{***} <0.01$.
(TIF)

**S5 Fig. Quantitative analysis of Fig 4. (A)** Quantitative analysis of protein level of SHP2 in human endometrial stromal cells (hESC) during decidualization progress. $^*$ indicated

experimental group (24h, 96h and 144h) versus control group (0h); # indicated experimental group (144h) versus experimental group (96h). **(B)** Quantitative analysis of SHP2 mRNA level in decidualization hESC. * indicated experimental group (24h, 96h and 144h) versus control group (0h); # indicated experimental group (144h) versus experimental group (96h); $ indicated experimental group (96h) versus experimental group (24h). **(C)** Quantitative analysis of protein expression of SHP2 in hESC stable knockout SHP2. **(D)** Quantitative analysis of protein expression of SHP2 in hESC stable overexpressed SHP2. The data are presented as the mean ± SD from three independent experiments. Statistical differences are indicated as follows: *P <0.05; **,##,$ $P <0.01.
(TIF)

**S6 Fig. The heatmap of different genes caused by Shp2 knockout at different time points in the process of induced decidualization of human endometrial stromal cells.** The threshold of differential genes is that any data conforms to | log2 (FoldChange) |> 0&padj <0.05. X axis is the sample cluster, and Y axis is the genes cluster. Red represents up-regulated gene transcription and blue represents down-regulated gene transcription. Color change shows the value of log2 (FoldChange).
(TIF)

**S7 Fig. The volcano map of different genes caused by Shp2 knockout at different time points in the process of induced decidualization of human endometrial stromal cells.** The threshold of differential genes is that any data conforms to | log2 (FoldChange) |> 0&padj <0.05. X axis is the value of log2 (FoldChange), and Y axis is -log10(padj). Red represents up-regulated gene transcription and blue represents down-regulated gene transcription.
(TIF)

**S8 Fig. Primary stromal cells isolated from day 4 pregnant uteri. (A)** The expression of Shp2 in primary stromal cells checked with immunofluorescence staining (n = 6). Green is positive color. **(B)**Purity assay of stromal cells with the marker protein vimentin (stromal cell marker) and cytokeratin (epithelial cell marker) labeled by immunofluorescence staining (n = 6). The cell nucleus was stained with DAPI. Scale bar: 100μm.
(TIF)

**S9 Fig. The enriched signal pathway analyzed by KEGG of different genes caused by Shp2 knockout at different time points in the process of induced decidualization of human endometrial stromal cells.** X axis is padj value and The Y-axis is enriched signal pathways.
(TIF)

**S10 Fig. Quantitative analysis of Fig 6. (A)** Quantitative analysis of immunohistochemical phospho-Erk1/2 in the D6 and D7 IS of Shp2^{f/f} and Shp2^{d/d} mice. **(B)** Quantitative analysis of protein level of Shp2, the ratio of phospho-Erk1/2 to total Erk1/2 proteins from D6 and D7 IS in Shp2^{f/f} and Shp2^{d/d} mice. **(C)** Quantitative analysis of protein level of Shp2, the ratio of phospho-Erk1/2 to total Erk1/2 proteins, and the ration of p-C/Ebpβ to C/Ebpβ in mouse primary stromal cells treated with the decidual regimen for 24 h. **(D)** Quantitative analysis of protein level of Shp2, the ratio of phospho-Erk1/2 to total Erk1/2 proteins in decidual induced hESC with or without SHP2. **(E)** Quantitative analysis of protein level of Shp2, the phospho-AKT to total AKT proteins, and the ration of p-FOXO1 to FOXO1 in decidual hESC. **(F)** Quantitative analysis of protein level of Shp2, phospho-STAT3, and total STAT3 in decidual hESC. The data are presented as the mean ± SD from three independent experiments. Statistical differences are indicated as follows: *P <0.05; **P <0.01.
(TIF)

**S11 Fig. The schematic flow chart about Shp2-mediated signal cascades including ERK-C/ EBPβ, AKT-FOXO1, and JAK-STAT3.**
(TIF)

**S1 Table. Primary antibody used in the assay of immunohistochemistry, Immunofluorescence and western blotting.**
(DOCX)

**S2 Table. Primer sequences used for realtime PCR.**
(DOCX)

**S1 Data. Raw data of 5891 differentially expressed genes (DEGs) identified based on a standard log2 (fold change)| > 0 and a padj <0.05.**
(XLSX)

## Author Contributions

**Conceptualization:** Zhongxian Lu.

**Data curation:** Jia Liang, Yingzhe Li, Xia Gao, Mengying Liu.

**Formal analysis:** Jianghong Cheng, Yingzhe Li, Wenbo Deng, Shuangbo Kong.

**Funding acquisition:** Haibin Wang, Shuangbo Kong, Zhongxian Lu.

**Investigation:** Jianghong Cheng, Jia Liang, Yingzhe Li, Xia Gao.

**Methodology:** Mengjun Ji, Yingpu Tian.

**Project administration:** Haibin Wang, Shuangbo Kong, Zhongxian Lu.

**Resources:** Gensheng Feng.

**Software:** Wenbo Deng.

**Supervision:** Haibin Wang, Shuangbo Kong, Zhongxian Lu.

**Validation:** Yingpu Tian.

**Writing – original draft:** Jianghong Cheng, Jia Liang, Shuangbo Kong, Zhongxian Lu.

**Writing – review & editing:** Haibin Wang, Shuangbo Kong, Zhongxian Lu.

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
