## [Decision Letter · Decision Letter 0]

11 Oct 2021

Dear Dr Lu,

Thank you very much for submitting your Research Article entitled 'Shp2 in uterine stromal cells critically regulates endometrial epithelial remodeling and stromal decidualization by multiple pathways during early pregnancy' to PLOS Genetics.

The manuscript was fully evaluated at the editorial level and by independent peer reviewers. The reviewers appreciated the attention to an important topic but identified some concerns that we ask you address in a revised manuscript

We therefore ask you to modify the manuscript according to the review recommendations. Your revisions should address the specific points made by each reviewer.

[LINK]

Yours sincerely,

Bruce Daniel Murphy

Guest Editor

PLOS Genetics

Gregory Barsh

Editor-in-Chief

PLOS Genetics

Your manuscript was favorably evaluated by two reviewers. Most of their comments can be addressed by clarification and rewrite. Two matters have been flagged by reviewer 2, including a better description of the phenotype and evaluation of the steroid profiles of the knockout vs WT animals.

Reviewer's Responses to Questions

**Comments to the Authors:**

Reviewer #1: General Comments: This paper from Liang and colleagues is a continuation of their investigation into the role of Shp2 in female reproductive function. Utilizing mouse genetics and cell culture, the authors show the cell type-specific functions of Shp2 during the establishment of pregnancy and demonstrate a potentially conserved role of Shp2 in human and mouse decidualization. Notably, the manuscript moves beyond descriptive analysis and provides a potential mechanism underlying defective decidualization in the absence of Shp2. Overall, this manuscript is a thorough report of the functions of Shp2 in the uterus during the establishment of pregnancy and will be of interest to female reproductive biologists. The minor comments I have are outlined below.

Specific comments:

The knockout efficiency of Amhr2Cre is known to be variable in different regions of the female reproductive tract (mesometrial vs. antimesometrial). Given that Shp2 is expressed throughout the stroma on day 6 of pregnancy (2017 PNAS paper), the authors should provide whole implantation site images of Shp2 IHC for control and knockout mice on day 6 in figure one.

In line with the previous comment, the authors should provide a lower magnification of the image presented in figure 2H to demonstrate the change in PTGS2 localization better.

Adding a schematic flow chart to Figure 6 would help the reader follow the presented results and mechanism proposed in this figure.

Implantation site weight is referred to in line 190 but is not shown in the figure.

It is important to clarify in the methods and figure legends if the hESC are primary or immortalized cells.

Reviewer #2: It is estimated that large numbers of pregnancies are failed due to blastocyst implantation failure or defects. The process of pregnancy failure still needs debate. Herein, the authors made conditional deletion of the Shp2 gene from the mouse uterine cells using Amrh2-Cre and this led to the defers the blastocyst implantation & inhibit the decidualization, leading to embryonic development delay & lethality of the embryo during midgestation, reducing the female fertility.

The lack of Shp2 in the stromal cells led to increased epithelial cell proliferation. In the HESC, Shp2 expression was increased & promoted decidual cell differentiation. Shp2 mediated signaling through ERK, AKT, STAT3, CCAAT, enhancer binding Protein β (CEBPβ), FOXO-1 during the decidualization.

The study is interesting and encompasses some critical findings. However, the study presentation is not precise and clear. Hence, the following recommendation is made.

1. The generation of the conditional knockout of Shp2 is not clearly defined and the authors need to demonstrate the other endocrine or reproductive function associated organs having a deletion of Shp2.

2. Line no. 162-165 and Fig S.1. The densitometric analysis, figure 1 A-S, is unclear.

3. Line no. 165-167. To know the differential expression of Shp2, a large area or rather full uterine tissue image is required.

4. What is the reason for the authors to study the expression level of Shp2 at Day12 rather than Day6?

5. How was the ovulation/ ovulated eggs as earlier images, it's not clear whether they have any difference in wild type than the wild type?

6. The serum levels of P2&E2 during the pregnancy failed either normal P2 &E2 or Shp2 downstream signaling.

7. The six months’ pregnancy is unclear and how many reproductive cycles have been covered /achieved, a detail of it is required.

8. Whether the eggs of both wild type and null have any visual or other difference is not clear.

9. The serum level of the P4 and E2 during the pregnancy period is suggested to be examined to know the hormonal and Shp2 cross-talk if any.

10. Line no. 222. It must be reduced proliferation, rather than attenuation.

11. Line nos. 225-227. Please confirm by epithelial biomarkers to claim that Epithelial cells are retained & assess from different replicates.

12. Line no.232. Why cox2 is having studied now any better rationale for that?

13. Line nos. 263-266. The quantitative analysis of Dtprp by staining intensity analysis as well as immunoblot / PCR is required to validate the given author's statement.

14. Line 261. The statement about the “defective differentiation of uterine stromal cells” is inappropriate as the authors did not assess the differentiation process either in vitro or in vivo.

15. Figure 4 & Lines 303-303. The Shp2 protein &mRNA levels were not quantified: hence, quantitative/ relative expression in fig 4. A, B, C, F, S, and D, can not be stated. Likewise in figure- 6 B, C, E, D, and G.

16. Figure 6. The fluorescence intensity value calculation is not done and visually the expression can not be appropriate to show semi-quantitative/comparative.

17. Line386. The statement has to be rephrased according to the localization pattern as it happened to be restricted to the epithelial region around the trophoblast site.

18. Figure 6 B, C, and D. Please explain the double band of ERK probed blot.

19. Figure 6F. FOXO1 double band and followed the trend to be explained.

20. Figure 6. The ERK, etc. expression or phosphorylation can also be linked to decidualization. However, the author's statement about the Shp2 dependent ERK signaling needs to be experimentally explained using normal stromal cells post Shp2 gain and loss functions then the Shp2 dependent ERK regulation can be stated.

21. Line 434. The Shp2 is not only ablated from the stromal cells as discussed.

22. A language check is suggested.

**Have all data underlying the figures and results presented in the manuscript been provided?**

Reviewer #1: Yes

Reviewer #2: Yes

PLOS authors have the option to publish the peer review history of their article (what does this mean?). If published, this will include your full peer review and any attached files.

Reviewer #1: No

Reviewer #2: **Yes: **Rajesh Kumar Jha

---

## [Decision Letter · Decision Letter 1]

27 Nov 2021

Dear Dr Lu,

Thank you very much for submitting your Research Article entitled 'Shp2 in uterine stromal cells critically regulates endometrial epithelial remodeling and stromal decidualization by multiple pathways during early pregnancy' to PLOS Genetics.

The manuscript was fully evaluated at the editorial level and by independent peer reviewers. The reviewers appreciated the attention to an important topic but identified some concerns that we ask you address in a revised manuscript

We therefore ask you to modify the manuscript according to the review recommendations. Your revisions should address the specific points made by each reviewer.

[LINK]

Yours sincerely,

Bruce Daniel Murphy

Guest Editor

PLOS Genetics

Gregory Barsh

Editor-in-Chief

PLOS Genetics

This manuscript is almost ready to go. Please have a look at Reviewer 2 comments related to typos and the lack o labelling of the axes of one of the supplementary figures (S9). The comment on line 232 may be pertinent as well, and some clarifications may be required. While the suggested further work would be interesting, it is not judged to be necessary.

Reviewer's Responses to Questions

**Comments to the Authors:**

Reviewer #1: The authors have adequately addressed my previous comments and the manuscript is suitable for publication.

Reviewer #2: Thanks for the opportunity to re-review the revised manuscript. The authors have answered the query to a certain extent and thankful to adhering and still, there is some clarification and supplemental data generation requirement as per the manuscript data presentation, and those are appended below.

The present Manuscript seems to be an extension of the already published article ‘Nuclear Shp2 directs normal embryo implantation via facilitating the ERα tyrosine phosphorylation by the Src kinase. From the same groups such as expression analysis at d4, and 5, the impact of Shp2 deletion on the embryo implantation, proliferation studies, ovulation, litter size, hormones levels, mucin expression level, etc. seemed to be overlapping, please look at these point very carefully.

The title is vague as per the presented data set and there are no remodeling studies neither the epithelial and epithelial cross-talk studies. Hence, the title needs to reflect the generated data and not be ambiguous.

The detail of the generation of the conditional knockout of Shp2 is not explained in the main manuscript text.

Lines 85-86 are overstated about the treatment of pregnancy failure.

If that is so then the authors can show the rescue experiment for Shp2 where the embryo implantation is studied in presence of recombinant shp2 supplementation in the shp2 -/- animals.

To prove further, the authors can generate the pregnancy failure mouse models and see the role of shp2.

Lines 206-208. Figures-1 C. PCR, Quantitation is necessary to know the relative change rather than just relying on the visual impression.

Line 209. There is no supporting data that AMHR-2 expression cells have SHP2 deletion.

Lines 212-213. The rationale to observe every 3 days is not clear as the gestation period is 21 and the estrous cycle s 4-5 days. How much gestation/pregnancy is seen during 6 months study?

Over the 06 months, each pregnancy outcome is to be analyzed separately rather than cumulative.

Lines 221-222. There is a lack of supporting data.

Fig. S9. It can be explained in detail as there is no caption for the x and y-axis.

Lines 232-233. The ovulation does not differ between the +/+ and -/- genotypes; hence, the statement might change accordingly.

Lines 234-236. The serum levels E2 and P4 and not largely affected and there is no statistical significance, which can confirm the same.

Line 232. The ‘02:00’ units are to be clarified.

Line 246. What is the status of the null mouse embryos on day5, 1000 h?

Line 251-254. It would ideally be good to study the cell proliferation markers in the null mouse.

Fig. 2F. How much area of tissue has been used to calculate the % change in Ki67 staining?

Lines 296-297. The data to support the area of IS is not presented to deduce any information.

In each of the experimental figure /result sections, mentioning the replicate's number will be more clear to the reader.

Line 362. Please look at the ‘padj’.

Figure-S6. It needs a better presentation to be clear to the reader rather than just only a heat map.

Likewise, figure-7 needs a clear presentation of the genes found altered.

Line 520 , 581, 1108, etc. Please correct the typo error.

**Have all data underlying the figures and results presented in the manuscript been provided?**

Reviewer #1: Yes

Reviewer #2: Yes

PLOS authors have the option to publish the peer review history of their article (what does this mean?). If published, this will include your full peer review and any attached files.

Reviewer #1: No

Reviewer #2: **Yes: **Rajesh Kumar Jha

---

## [Editor Report · Decision Letter 2]

5 Jan 2022

Dear Dr Lu,

We are pleased to inform you that your manuscript entitled "Shp2 in uterine stromal cells critically regulates on time embryo implantation and stromal decidualization by multiple pathways during early pregnancy" has been editorially accepted for publication in PLOS Genetics. Congratulations!

Yours sincerely,

Bruce D. Murphy

Guest Editor

PLOS Genetics

Gregory Barsh

Editor-in-Chief

PLOS Genetics

Comments from the reviewers (if applicable):

The authors have adequately addressed the issues raised by reviewer 2.

**Data Deposition**

http://datadryad.org/submit?journalID=pgenetics&manu=PGENETICS-D-21-01259R2

**Press Queries**

---

## [Editor Report · Acceptance letter]

11 Jan 2022

PGENETICS-D-21-01259R2 

 Shp2 in uterine stromal cells critically regulates on time embryo implantation and stromal decidualization by multiple pathways during early pregnancy 

Dear Dr Lu, 

We are pleased to inform you that your manuscript entitled " Shp2 in uterine stromal cells critically regulates on time embryo implantation and stromal decidualization by multiple pathways during early pregnancy" has been formally accepted for publication in PLOS Genetics! Your manuscript is now with our production department and you will be notified of the publication date in due course.

With kind regards,

Zsofia Freund

PLOS Genetics

On behalf of:
